# ADVANCING SUPERVISED LOCAL LEARNING BEYOND CLASSIFICATION WITH LONG-TERM FEATURE BANK

## ABSTRACT

Local learning offers an alternative to traditional end-to-end back-propagation in deep neural networks, significantly reducing GPU memory consumption. Although it has shown promise in image classification tasks, its extension to other visual tasks has been limited. This limitation arises primarily from two factors: 1) architectures designed specifically for classification are not readily adaptable to other tasks, which prevents the effective reuse of task-specific knowledge from architectures tailored to different problems; 2) these classification-focused architectures typically lack cross-scale feature communication, leading to degraded performance in tasks like object detection and super-resolution. To address these challenges, we propose the Feature Bank Augmented auxiliary network (FBA), which introduces a simplified design principle and incorporates a feature bank to enhance cross-task adaptability and communication. This work presents the first task-agnostic framework that extends supervised local learning beyond classification to a broad range of visual tasks, demonstrating that FBA not only conserves GPU memory but also achieves performance on par with end-to-end approaches across multiple datasets for various visual tasks.

## 1 INTRODUCTION

Back-propagation (BP) remains the cornerstone of deep learning optimization, but as models scale to larger sizes Bengio et al. (2006); Krizhevsky et al. (2017), End-to-End (E2E) methods expose several limitations Hinton et al. (2006); Guo et al. (2020). BP relies on the propagation of error signals across multiple layers, a process that contrasts with biological neural transmission systems Crick (1989) and introduces challenges, such as error accumulation in deep networks. This can degrade the learning effectiveness of shallow neurons Qu et al. (1997). Moreover, updating hidden layers in the deep network requires the completion of forward and backward passes, which hinders parallel computation and significantly increases memory consumption on GPUs Jaderberg et al. (2017); Belilovsky et al. (2020). As an alternative to E2E methods, supervised local learning enhances memory efficiency and parallelism by splitting the network into gradient-isolated blocks, each updated independently via its own auxiliary network Belilovsky et al. (2020); Nøkland & Eidnes (2019).

However, current applications of local learning have largely been confined to image classification tasks, where they have demonstrated competitive performance Ma et al. (2024); Wang et al. (2021) compared to End-to-End (E2E) methods through tailor-made auxiliary networks. Despite these successes, the focus on auxiliary network architectures for classification has limited their broader applicability. When extending these architectures to more complex tasks, such as pixel-wise task, they often fall short. This limitation arises due to their lack of cross-task adaptability and the widely recognized "short-sightedness" problem Su et al. (2024b).

Although the work Su et al. (2024a) alleviates short-sightedness by using exponential moving averages to enhance single-scale communication, it fails to address deeper limitations stemming from cross-task adaptability challenges—particularly in scenarios where the model must process information at different scales to meet the diverse requirements of various tasks. For instance, high-level tasks often rely on contextual representations across broader scales, whereas low-level tasks demand fine-grained, pixel-level information. The classification-oriented architecture's inherent lack of such

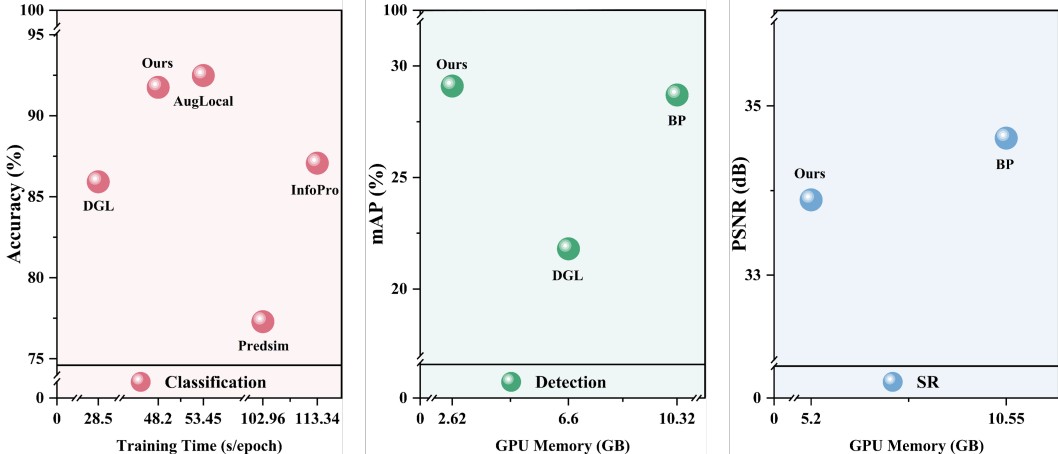

Figure 1: Performance Comparison of FBA Across Multiple Tasks. (a) In classification tasks, FBA is compared with state-of-the-art local learning methods in terms of training speed and accuracy. (b) and (c) For object detection and super-resolution, FBA is evaluated against back-propagation (BP) with respect to GPU memory overhead and accuracy. For a detailed analysis of GPU memory usage in classification tasks, please refer to the supplementary materials.

scale diversity further exacerbates the issue of short-sightedness. As a result, these limitations constrain the potential of traditional local learning methods, hindering their generalization and transferability across a wide range of visual tasks, and methods such as MAN and Infopro must substantially modify their architectures—introducing complex task-specific local-learning modules—whenever extending to new tasks.

To this end, we present the Feature Bank Augmented auxiliary network (FBA), a novel framework designed to address the challenges of scaling local learning methods across diverse tasks. This streamlined approach alleviates the above short-sightedness issue between local modules at different scales and enables performance that closely matches end-to-end training. Specifically, FBA operates as an auxiliary network within each gradient-isolated local module, adapting automatically to the target task's architecture, thus eliminating the need for manual design adjustments. It features a straightforward local module that emphasizes the reusability of task-specific knowledge, facilitating the extension of local learning to various. A key innovation is the incorporation of a feature bank to enable multi-scale feature communication, allowing FBA to capture both generalized and discriminative semantic features. By integrating cross-scale information from the feature bank, FBA constructs a comprehensive semantic representation, advancing supervised local learning beyond classification tasks. Extensive experiments show that FBA achieves comparable performance to end-to-end methods on a variety of challenging tasks shown in Figure1, including image classification, object detection, and super-resolution, while significantly saving GPU memory.

The main contributions of this paper are as follows:

- This paper introduces the Feature Bank Augmented auxiliary network (FBA), which simplifies the network structure for corresponding tasks. By facilitating access to cross-scale features, it effectively addresses the needs of diverse applications, enabling the seamless extension of local learning.

- Comprehensive experiments on image classification, object detection, and super-resolution tasks validate the effectiveness of the FBA designed local learning network. The FBA approach achieves performance comparable to end-to-end back-propagation (BP) while significantly reducing GPU memory usage.

- An in-depth analysis of the latent representations learned by models utilizing the FBA method reveals that, compared to BP, local networks enhanced with key global information help the network learn more discriminative features at shallow layers, thereby improving the overall performance of the model.

## 2 RELATED WORK

We briefly review local learning and related alternatives to end-to-end (E2E) training. For detailed task-specific discussions, please see supplementary material.

### 2.1 ALTERNATIVE METHODS TO E2E TRAINING

Increasingly evident limitations of E2E training have led researchers Lillicrap et al. (2020); Crick (1989); Nøkland (2016); Clark et al. (2021); Lillicrap et al. (2016); Akrout et al. (2019) to explore biologically plausible alternatives. Recent approaches Ren et al. (2022); Dellaferrera & Kreiman (2022) employed forward gradient learning to circumvent backpropagation drawbacks, enhancing biological plausibility but struggling with large-scale dataset performance Deng et al. (2009). Moreover, their reliance on global objectives remains fundamentally misaligned with real-world neural structures, which depend primarily on local neuron connections.

### 2.2 LOCAL LEARNING

Local learning methods Crick (1989) improve memory efficiency and address limitations of global E2E training Hinton et al. (2006) by utilizing supervised local losses or auxiliary networks. Previous works incorporated self-supervised contrastive losses under local constraints Illing et al. (2021); Xiong et al. (2020); Nøkland & Eidnes (2019); Wang et al. (2021), enabling block-level training through manually selected auxiliary networks Pyeon et al. (2020); Belilovsky et al. (2020). However, dividing networks into local blocks induces a "short-sightedness" issue, limiting parameter communication across blocks. Su et al. Su et al. (2024a; 2025); Guo et al. (2024) mitigated this using exponential moving averages but introduced substantial computational overhead due to frequent memory access. Additionally, previous approaches primarily targeted semantic classification, failing to generalize effectively to tasks requiring different scales or pixel-level precision. In contrast, our FBA method addresses these challenges by explicitly maintaining key task-specific features in a Feature Bank, significantly reducing memory writes and accelerating local learning. Explicitly preserved features further facilitate performance across diverse visual tasks, proving essential for real-world industrial applications Rath & Condurache (2024); Zhang et al. (2024).

## 3 METHOD

### 3.1 PRELIMINARIES

We first briefly revisit traditional end-to-end (E2E) supervised learning and local learning methods to clarify our motivation.

In standard E2E supervised learning, a deep network is partitioned into multiple sequential blocks, each parameterized by $\theta_j$. Given an input sample $x$ and its ground-truth label $y$, the network generates predictions via forward propagation $x_{j+1} = f_{\theta_j}(x_j)$. The final output $\hat{y}$ is compared against $y$ using a loss function $\mathcal{L}(\hat{y}, y)$, whose gradients propagate backward through all blocks.

Local learning approaches Nøkland & Eidnes (2019); Wang et al. (2021); Belilovsky et al. (2020) simplify this process by introducing auxiliary networks for local supervision. Each local block $j$ has an auxiliary network with parameters $\gamma_j$ generating local predictions $\hat{y}j = g\gamma_j(x_{j+1})$. Parameters are locally updated by:

$$\gamma_j \leftarrow \gamma_j - \eta_a \nabla_{\gamma_j} \mathcal{L}(\hat{y}j, y), \quad \theta_j \leftarrow \theta_j - \eta_l \nabla \theta_j \mathcal{L}(\hat{y}_j, y), \tag{1}$$

where $\eta_a, \eta_l$ denote learning rates of auxiliary and local networks, respectively. Such local supervision enables gradient isolation, reducing complexity.

However, existing local learning methods face two critical limitations when applied across tasks. First, auxiliary networks are typically task-specific, limiting generalizability. Second, local modules suffer from short-sightedness due to limited receptive fields, impeding transfer to tasks demanding richer features Su et al. (2024b;a).

To address these issues, we propose the Feature Bank Augmented auxiliary network (FBA) framework, designed explicitly to generalize local learning effectively to multiple tasks.

## 3.2 ARCHITECTURE

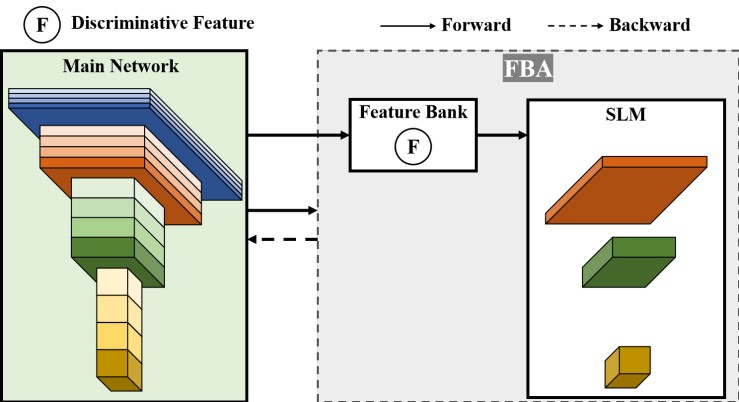

Figure 2: Structural diagram of the FBA method. FBA consists of a Feature Bank and SLM. Discriminative features are extracted from the primary network, and an SLM, designed to be homogeneous with the backbone, is constructed so that both the local modules and the backbone collaboratively update the gradients.

We proposed Feature Bank Augmented auxiliary network (FBA) architecture generalizes local learning across diverse vision tasks. FBA contains two key components: Simple Local Modules (SLM) and a Feature Bank, as depicted in Figure 2.

**Simple Local Modules (SLM).** Unlike previous task-specific auxiliary networksBelilovsky et al. (2020); Wang et al. (2021), SLMs are designed to be both simple and generalizable. We first partition the backbone $\mathcal{B}$ into $K$ local modules. Each module corresponds to an internal block $b_i$ ($i \in \{1, \ldots, K\}$), which can be located at any resolution or depth. To build the auxiliary network $\mathcal{A}_i$, we *directly reuse the **last block** of $b_i$'s current stage*, and append a original task head:

$$\mathcal{A}_i = \underbrace{\phi\big(b_i^{\text{last}}\big)}_{\text{Simplified block}} + \underbrace{\text{Head}_{\text{task}}}_{}, \tag{2}$$

where $b_i^{\text{last}}$ denotes the last residual unit inside $b_i$'s stage, and $\phi(\cdot)$ is a shallow "pass-through" operator that optionally reduces channels while preserving kernel size and stride. $\text{Head}_{\text{task}}$ is the main task head . This *stage-tail reuse* keeps critical feature extraction capacity and guarantees architectural alignment with the backbone, while keeps each SLM cheap enough for independent local training.

**Feature Bank.** To compensate for the short receptive field of local learning, we cache task-critical backbone features in a Feature Bank $\mathcal{F}_{\text{bank}}$ during training and feed them to each auxiliary network on demand. Concretely, we store GAP/FC activations for classificationNøkland & Eidnes (2019), key multi-scale maps for detectionLin et al. (2017a), and early full-resolution maps for super-resolutionLim et al. (2017); Tian et al. (2024).

The Feature-Bank-augmented forward of an auxiliary module is:

$$f_{\mathcal{A}i}(\mathbf{x}) = \text{Headtask}\left(\text{Simplify}(b_i(\mathbf{x})) \oplus \mathcal{F}_{\text{bank}}^{(i)}\right), \tag{3}$$

where $\oplus$ is a light fusion and $\mathcal{F}_{\text{bank}}^{(i)}$ is the slice relevant to block$b_i$. The bank is *training-only*; inference remains identical to the original backbone, incurring zero extra cost.

Figure 3: FBA application across tasks. Dashed lines indicate gradient feedback flow. Subnetworks below depict layer-specific local modules, with colors denoting distinct features. Top: object detection; bottom: classification.

# 4 INSTANTIATIONS

To demonstrate the generalizability of FBA, we instantiate it on three representative computer vision tasks: image classification, object detection, and super-resolution. These tasks respectively represent traditional local learning settings, high-level structured prediction tasks, and low-level dense reconstruction tasks. Unlike conventional end-to-end pipelines where task adaptation is achieved by replacing the head, local learning poses additional challenges due to gradient isolation and incomplete feature flow.

FBA addresses this issue through two design choices: (1) each auxiliary module (SLM) reuses simplified backbone blocks and the task-specific head, and (2) the Feature Bank supplements the local receptive field with key global or task-relevant features. Below, we illustrate how FBA is instantiated in classification networks; detection and super-resolution variants follow similar design with task-specific considerations.

## 4.1 INSTANTIATION IN CLASSIFICATION

We take ResNet as the representative classification backbone, as shown in Figure 3 (bottom). Let the backbone network $\mathcal{B}$ be divided into $K$ local modules, each corresponding to an internal block or group of layers. For each local module indexed by $i \in 1, 2, ..., K$, we construct a corresponding auxiliary network $\mathcal{A}_i$ with the following structure:

$$\mathcal{A}_i = \text{SLM}(b_i(\cdot), \mathcal{F}_{\text{bank}}) + \text{ClsHead}, \tag{4}$$

where $b_i$ is the $i$-th selected block from the backbone $\mathcal{B}$, Simplify$(\cdot)$ denotes a lightweight transformation (e.g., reducing depth or channel size), and TaskHead reuses a simplified version of the original task-specific head (e.g., classifier, detector, or reconstructor).

Each $\mathcal{A}i$ is trained independently using local supervision based on the same loss structure as the main task, enhanced by task-critical features provided by the Feature Bank $\mathcal{F}$bank. Compared with stage-level auxiliary structures Belilovsky et al. (2020); Wang et al. (2021), this fine-grained block-level design allows greater flexibility and finer control over local learning.

## 4.2 INSTANTIATION IN OBJECT DETECTION

Object detection presents additional challenges for local learning due to its heavy reliance on multi-scale features and dense spatial reasoning. We instantiate FBA in a typical FPN-based detector , as illustrated in Figure 3 (top).

Same as we do in classification, we divide the backbone into $K$ local modules and construct an SLM by simplifying the corresponding block and appending a lightweight detection head , ensuring alignment with the global task objective.

Since FPN-based detectors aggregate features from multiple scales, we enhance each SLM with cross-scale information via the Feature Bank. Specifically, during training, we store multi-scale feature maps from the backbone's key downsampling points into $\mathcal{F}_{\text{bank}}$ . Each SLM then uses these stored features to construct a lightweight local FPN, enabling it to generate predictions with sufficient scale context:

$$f_{\mathcal{A}i}(\cdot) = \text{LocalFPN} \left( b_i(\cdot), \mathcal{F}\text{bank} \right) \tag{5}$$

This design avoids manually crafting task-specific auxiliary structures while ensuring that each local module is receptive to the multi-scale nature of object detection Lin et al. (2017a). The Feature Bank is only active during training and does not affect inference performance or latency.

### 4.3 INSTANTIATION IN SUPER-RESOLUTION

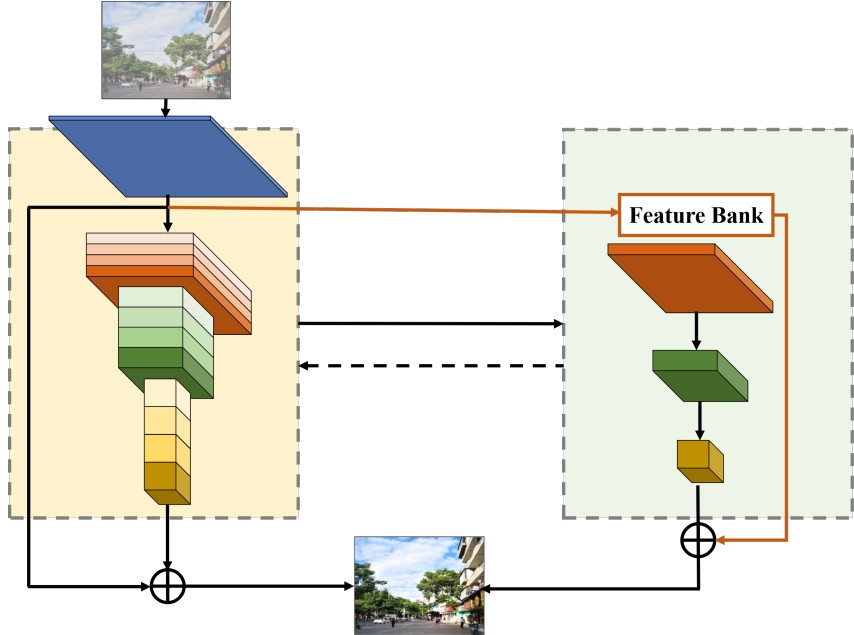

Figure 4: The structure diagram of FBA applied to sr shows the main network on the left and the corresponding local modules in the green boxes on the right. Key features are highlighted, and their flow is indicated by orange lines.

Super-resolution (SR) requires dense pixel-wise reconstruction, making it highly sensitive to spatial detail and full-resolution information. To instantiate FBA for SR, as illustrated in Figure 4, we follow a similar strategy as in classification and detection: construct an SLM by simplifying the associated block and appending a lightweight reconstruction head, aligned with the original SR objective.

Unlike detection tasks that rely on multi-scale aggregation, SR models often depend on preserving and refining full-resolution features throughout the network Lim et al. (2017); Tian et al. (2024); Chen et al. (2023). To address this, we store early-stage high-resolution feature maps in the Feature Bank $\mathcal{F}_{\text{bank}}$ during training. These features are then reused by each SLM to recover fine-grained details otherwise lost due to limited local receptive fields.

Formally, each auxiliary module predicts a super-resolved output using:

$$f_{\mathcal{A}i}(\cdot) = \text{SRHead} \left( b_i(\cdot), \mathcal{F}\text{bank} \right), \tag{6}$$

Table 1: Comparison of classification performance on four datasets. We report Top-1 accuracy (ACC) and per-epoch training time for CIFAR-10, STL-10, SVHN, and ImageNet, comparing our method with state-of-the-art approaches. The symbol "–" denotes results not reported in the original papers.

| Method | CIFAR-10 | | STL-10 | | SVHN | | ImageNet | |
|---|---|---|---|---|---|---|---|---|
| | ACC | Training Time | ACC | Training Time | ACC | Training Time | ACC | Training Time |
| Predsim | 77.29 | 102.96s | 67.10 | 111.75s | 91.92 | 102.84s | – | – |
| DGL | 85.92 | 28.50s | 72.86 | 20.24s | 94.95 | 25.72s | – | – |
| InfoPro | 87.07 | 113.34s | 70.72 | 107.73s | 94.03 | 116.64s | 78.15 | 58min |
| AugLocal | 92.48 | 53.45s | 79.69 | 79.06s | 96.80 | 65.96s | 78.70 | 150min |
| Ours | 91.75 | 48.20s | 79.74 | 75.05s | 96.68 | 53.10s | 78.81 | 67min |

where SRHead($\cdot$) denotes the lightweight reconstruction layers reused from the global model's head, and $b_i$ is the simplified local block. The Feature Bank ensures each local module receives fine spatial guidance, improving edge sharpness and texture recovery without introducing inference-time overhead.

## 5 EXPERIMENTS

We evaluate FBA on three tasks: classification, detection, and super-resolution. For classification, we use CIFAR-10 Krizhevsky et al. (2009), SVHN Netzer et al. (2011), STL-10 Coates et al. (2011), and ImageNet Deng et al. (2009), comparing against four state-of-the-art supervised local learning methods: PredSim Nøkland & Eidnes (2019), DGL Belilovsky et al. (2020), InfoPro Wang et al. (2021), and AugLocal Ma et al. (2024). For detection, we conduct quantitative evaluations on the VOC Everingham et al. (2010) and COCO Lin et al. (2014) datasets. Our comparisons include both traditional end-to-end detection methods and advanced local learning approaches, with particular attention to accuracy and memory overhead. Notably, conventional detection methods typically use ImageNet-pretrained weights for initialization, whereas no such pre-trained weights exist for local learning. To ensure a fair comparison, all detection models are trained from scratch with random initialization. This accounts for the discrepancies observed in Table 3, where the BP model results differ from the numbers reported in prior studies. For the super-resolution task, we validate the performance of FBA on the widely used DIV2K Agustsson & Timofte (2017) dataset.

### 5.1 IMPLEMENTATION DETAILS

The experiments on classification were conducted on an NVIDIA A100 80GB GPU using the SGDKeskar & Socher (2017) optimizer with a cosine annealing learning rate schedule. The initial learning rate was set to 0.8, and the model was trained for 400 epochs. The experiments on object deticion,We use the SGD optimizer with Nesterov momentumDozat (2016) (0.9) and a weight decay of 1e-4. Training lasts 90 epochs with a cosine annealing learning rate, preceded by a brief warm-up phase. And in super resolution, we use 48×48 low-resolution patches and corresponding high-resolution patches are used. ADAM optimizer is applied with a learning rate of 1e-4. Training starts from scratch on the ×2 model, which, after convergence, serves as a pre-trained network for ×3 and ×4 models. Further details of the experiment can be found in the supplementary materials.

### 5.2 EVALUATION IN CLASSIFICATION

We benchmark FBA on four classification datasets. For CIFAR-10, STL-10 and SVHN every method uses a ResNet-32 divided into K=16 blocks. On ImageNet we follow each baseline's published setting. As shown in Table 1, although achieving the best performance was not our primary goal, FBA still obtains the best or second-best results on each dataset, indicating that it maintains strong transferability without sacrificing performance. Notably, in terms of training speed, due to the simplicity of the SLM design, we avoid using complex auxiliary network structures, allowing

FBA to achieve remarkable speed, second only to DGL. However, DGL's accuracy is significantly lower than FBA's, highlighting FBA's superior balance between performance and efficiency.

## 5.3 EVALUATION IN OBJECT DETECTION

**Results on VOC dataset:** To verify the performance of the FBA method, we first conduct experiments on VOC dataset using the traditional BP method with our FBA. The experimental results are shown in Table 2. Surprisingly, the FBA method achieves comparable performance to the BP method in the vast majority of experimental groups. It is worth noting that the FBA method achieves higher mAP results in the experiments with RetinaNet-R50 and other models. This improvement is attributed to our model scoring better on smaller objects such as bottles and cars. This may be due to the fact that the FBA method can help the model identify the focal features earlier, while the shallow layers used to identify small objects can effectively learn more discriminative features. This leads to an overall performance enhancement of the model.

Moreover, we observe that even though the mAP performance of the FBA method is comparable to that of the BP method, its $AP_{50}$ and $AP_{75}$ scores are still lower than those of the BP method. This suggests that at higher threshold Settings, the features learned by the FBA method may be more discriminative than the BP method, leading to better performance at these thresholds.

Table 2: Performance comparison on the VOC validation set.

| Model | mAP | aero | bike | bird | boat | bottle | bus | car | cat | chair | cow | table | dog | horse | mbike | person | plant | sheep | sofa | train | tv |
|---|---|---|---|---|---|---|---|---|---|---|---|---|---|---|---|---|---|---|---|---|---|
| RetinaNet-R34 | 53.9 | 68.6 | 58.2 | 49.4 | 40.3 | 21.5 | 58.2 | 50.7 | 81.6 | 31.9 | 51.0 | 43.8 | 76.9 | 71.5 | 65.3 | 54.8 | 22.4 | 43.2 | 55.6 | 76.6 | 57.3 |
| Ours (K=17) | 52.2 | 64.8 | 55.6 | 44.5 | 39.7 | 24.1 | 58.6 | 56.3 | 83.4 | 28.6 | 49.8 | 37.4 | 66.3 | 69.4 | 58.9 | 57.3 | 22.6 | 36.8 | 53.9 | 80.6 | 51.4 |
| RetinaNet-R50 | 56.2 | 67.5 | 59.6 | 53.1 | 44.8 | 24.1 | 58.2 | 54.5 | 82.6 | 30.7 | 57.8 | 44.5 | 80.9 | 76.8 | 68.3 | 56.5 | 21.2 | 46.5 | 57.9 | 79.1 | 60.2 |
| Ours (K=17) | 56.5 | 64.9 | 51.3 | 56.5 | 45.2 | 25.3 | 58.9 | 55.7 | 85.1 | 29.3 | 58.4 | 40.5 | 73.9 | 76.6 | 64.5 | 59.2 | 22.9 | 37.5 | 56.5 | 83.2 | 59.0 |
| RetinaNet-R101 | 58.2 | 69.9 | 61.6 | 53.0 | 51.5 | 26.1 | 61.7 | 57.1 | 84.3 | 35.5 | 58.7 | 44.4 | 81.4 | 77.1 | 70.4 | 58.7 | 24.0 | 45.8 | 61.7 | 81.8 | 60.8 |
| Ours (K=34) | 56.9 | 62.4 | 55.5 | 57.1 | 50.9 | 25.5 | 61.4 | 55.9 | 86.7 | 32.4 | 57.3 | 39.9 | 77.1 | 77.4 | 65.5 | 59.4 | 25.2 | 35.5 | 57.9 | 83.9 | 56.5 |
| RetinaNet-R152 | 61.0 | 72.2 | 64.8 | 57.7 | 50.2 | 31.9 | 62.8 | 59.9 | 85.3 | 41.2 | 63.2 | 53.3 | 81.9 | 78.9 | 70.2 | 61.2 | 28.5 | 47.4 | 63.2 | 81.5 | 65.0 |
| Ours (K=51) | 60.9 | 69.4 | 60.5 | 62.7 | 51.1 | 30.5 | 65.1 | 56.4 | 83.5 | 43.3 | 60.7 | 54.0 | 74.9 | 81.4 | 62.8 | 63.7 | 27.1 | 45.5 | 63.1 | 85.5 | 65.6 |
| YOLO-R34 | 58.9 | 63.6 | 65.2 | 62.9 | 42.2 | 30.6 | 67.7 | 67.4 | 77.3 | 36.4 | 63.5 | 49.9 | 74.3 | 76.8 | 67.5 | 60.6 | 27.4 | 60.0 | 52.2 | 72.9 | 60.3 |
| Ours (K=17) | 58.6 | 64.2 | 63.5 | 63.3 | 34.8 | 30.1 | 66.9 | 65.1 | 77.3 | 30.5 | 62.8 | 48.5 | 70.3 | 82.6 | 65.4 | 61.0 | 28.1 | 61.2 | 49.7 | 72.4 | 61.3 |
| YOLO-R50 | 58.5 | 57.0 | 73.2 | 60.9 | 37.8 | 30.4 | 66.6 | 66.5 | 76.7 | 37.7 | 61.1 | 44.2 | 76.9 | 77.0 | 67.8 | 60.1 | 29.3 | 58.8 | 56.5 | 64.6 | 60.8 |
| Ours (K=17) | 57.1 | 56.9 | 73.7 | 59.4 | 35.5 | 31.8 | 61.4 | 67.8 | 77.9 | 34.5 | 60.4 | 44.4 | 68.5 | 81.8 | 66.1 | 61.9 | 29.1 | 54.3 | 55.9 | 70.1 | 60.9 |
| YOLO-R101 | 60.4 | 65.1 | 68.1 | 64.9 | 45.1 | 27.5 | 69.1 | 68.2 | 76.7 | 38.6 | 65.0 | 51.1 | 76.6 | 78.8 | 73.4 | 62.5 | 32.4 | 62.2 | 57.2 | 67.7 | 57.1 |
| Ours (K=34) | 60.6 | 65.7 | 64.5 | 62.9 | 47.4 | 29.1 | 67.1 | 70.7 | 78.4 | 36.9 | 64.3 | 53.4 | 70.2 | 79.9 | 74.2 | 61.8 | 33.5 | 59.9 | 57.5 | 70.5 | 55.8 |

Table 3: COCO val results: ↑ shows FBA's accuracy gain over other local-learning methods, and ↓ indicates the memory saved versus BP baselines.

| Model | Method | $mAP$ | $AP_{50}$ | $AP_{75}$ | GPU Memory |
|---|---|---|---|---|---|
| RetinaNet-R34 | BP | 28.7 | 49.3 | 29.5 | 10.32GB |
| | DGL(K=4) | 21.8 | 44.6 | 17.2 | 6.60GB |
| | Ours(K=4) | 29.1(↑ 7.3) | 49.3(↑ 4.7) | 28.9(↑ 11.7) | 2.62GB (↓74.6%) |
| | w/o Feature Bank | 20.7 | 42.3 | 15.8 | 2.34GB |
| RetinaNet-R50 | BP | 29.7 | 51.6 | 30.4 | 18.05GB |
| | Ours(K=4) | 29.8 | 50.6 | 30.5 | 5.92GB (↓67.02%) |
| RetinaNet-R101 | BP | 31.8 | 53.6 | 32.3 | 22.46GB |
| | Ours(K=4) | 31.9 | 52.7 | 32.8 | 6.34GB (↓71.77%) |
| RetinaNet-R152 | BP | 33.2 | 56.2 | 33.5 | 31.84GB |
| | Ours(K=4) | 33.7 | 56.7 | 33.8 | 7.83GB (↓75.40%) |
| YOLO-R34 | BP | 20.23 | 41.27 | 21.10 | 11.80GB |
| | Ours(K=4) | 20.26 | 40.74 | 20.94 | 3.36GB (↓71.53%) |
| YOLO-R50 | BP | 20.94 | 42.02 | 21.97 | 26.15GB |
| | Ours(K=4) | 20.98 | 42.14 | 20.19 | 6.83GB (↓73.88%) |
| YOLO-R101 | BP | 22.41 | 44.36 | 22.53 | 37.05GB |
| | Ours(K=4) | 22.46 | 43.89 | 22.47 | 13.77GB (↓62.83%) |
| YOLO-R152 | BP | 25.00 | 47.15 | 24.33 | 49.88GB |
| | Ours(K=4) | 24.87 | 46.16 | 24.73 | 15.61GB (↓68.7%) |

**Result on MS COCO:** We extensively evaluate our proposed FBA method on the challenging MS COCO dataset Lin et al. (2014). To control experimental cost and considering the efficiency of DGL (Table 1), we first adopt the RetinaNet-R34 backbone for comparison.For a fair baseline, we augment DGL with an SLM auxiliary network equipped with an FPN-style multi-scale structure, so that DGL can be effectively applied to object detection.Our full method further integrates a Feature Bank on top of SLM to enhance cross-task feature reuse.As shown in Table 3, the SLM-enhanced

DGL still lags far behind standard BP (21.8 mAP vs 28.7 mAP). By contrast, our FBA achieves 29.1 mAP, and removing the Feature Bank drops performance to 20.7 mAP. These results indicate that SLM mainly provides the cross-task adaptability, while the performance gain is largely attributed to the Feature Bank.Furthermore, across deeper backbones, our method consistently matched or slightly outperformed the BP-based method, indicating the robustness and scalability of FBA.

It should be noted that, for experiments involving the YOLO framework, we specifically employed ResNet-based backbones instead of recent YOLO variants such as YOLOv8 or YOLOv9. This deliberate choice is primarily driven by our intention to maintain consistency in backbone architectures across all experiments, enabling fair comparison and alignment with traditional local-learning methods, which have been extensively validated on convolutional architectures such as ResNet and VGG.

When comparing GPU memory usage, FBA demonstrates superior memory-saving capabilities compared to BP due to our simplified local structure. In RetinaNet-R34, YOLO-R34, and RetinaNet-R101, FBA reduces memory overhead by 74.6%, 71.53%, and 71.77%, respectively, while maintaining comparable or better detection performance.

**Ablation Study:** We perform ablation experiments on FBA; due to space limitations, we only present part of the experiments in the main text, and other experiments will be provided in the supplementary material.

We try to incrementally reduce the modules of the local detection, and the results are shown in Table 4. Where Adapt represents whether to use FBA's SLM and Feature bank methods, and Head represents whether to share the same detection head with the network. We find that the shared detection head can help the model improve the performance at a certain increase in memory overhead. This shows that while one can simply introduce local learning methods to the task, there is still much room for improvement in how to exploit these important features once they are added to the local network. There may be potential to consistently outperform BP architectures. But achieving state-of-the-art performance in each task is not the main goal of this paper; We leave it as future work.

Table 4: Ablation study between modules of different local detection schemes. Here, Adapt indicates whether the adaptive FBA network is used, and Head indicates whether the shared prediction head is used.

| Adapt | Head | mAP | GPU Mem |
|-------|------|-----|---------|
| × | × | 28.7 | 10.32 |
| ✓ | × | 27.7 | 8.07 |
| ✓ | ✓ | 28.5 | 8.19 |

Table 5: Results on the validation set of DIV2K.

| Task | Method | PSNR | GPU Mem |
|------|--------|------|---------|
| ×2 | BP | 34.62 | 10.55GB |
| | Ours | 33.89 | 5.20GB |
| ×3 | BP | 31.04 | 10.30GB |
| | Ours | 29.33 | 5.19GB |
| ×4 | BP | 28.92 | 10.06GB |
| | Ours | 27.71 | 5.16GB |

### 5.4 Evaluation in Super-Resolution

Table 5 shows that our FBA method training cuts GPU memory usage by roughly 50% at all scaling factors, because global back-propagation is applied only to the backbone while each SLM is updated locally. The freed memory can be reinvested in larger batch sizes or higher-resolution patches during training. The corresponding PSNR reduction is limited—-0.73dB (×2), -1.71dB (×3), and -1.21dB (×4)—indicating that FBA retains most of the performance benefits of full BP while halving memory cost. Interestingly, the quality gap narrows at ×4, suggesting that FBA's Feature Bank helps capture the coarse-scale information increasingly dominant in harder up-scaling tasks.

## 6 Conclusion

We introduce Feature Bank Augmented auxiliary network (FBA), a novel auxiliary design that extends local learning to diverse tasks while achieving performance comparable to BP. FBA leverages the backbone network's structure and multi-level features, eliminating manual configuration. By augmenting feature memory and enhancing cross-scale information utilization within local modules, FBA significantly reduces GPU memory usage while maintaining BP-level performance on tasks like object detection and image super-resolution.

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

# A APPENDIX

## A.1 MORE RELATE WORK

### A.1.1 OBJECT DETECTION

R-CNN and Faster R-CNNGirshick et al. (2014); Ren et al. (2015) are the origin and excellent succession of the classic R-CNN model, respectively. They employed simple and scalable networks for object detection, yet achieved very high detection accuracy. YOLOv1Redmon et al. (2016) and YOLOv8Jocher (2023) represent the pioneering work and the latest iteration of the YOLO (You Only Look Once) series, respectively. They treat object detection as a regression problem to spatially separated bounding boxes and associated class probabilities, making it a real-time, fast object detection model. On the other hand, RetinaNetLin et al. (2017b) is a dense detector utilizing focal loss, offering high detection accuracy. DETRCarion et al. (2020) simplified the detection process by directly treating object detection as a set prediction problem. This significantly reduced the need for many components. However, the aforementioned methods still face the issue of high memory consumption during training.

### A.1.2 IMAGE SUPER-RESOLUTION

Image Super-Resolution (SR) research aims to reconstruct High-Resolution (HR) images from Low-Resolution (LR) images. This technology has significant applications in various fieldsWang et al. (2020); Georgescu et al. (2023); Razzak et al. (2023). SRCNNDong et al. (2015) is the pioneer of deep learning-based super-resolution models. It is a simple model that addresses the image restoration problem using just three layers, achieving impressive results. EDSRLim et al. (2017) is an enhanced deep super-resolution network that improves model performance by removing unnecessary modules from the traditional residual network. RCANZhang et al. (2018) and SwinIRLiang et al. (2021) utilize a very deep residual channel attention network and Swin Transformer, respectively, for high-precision image super-resolution. Both have achieved outstanding results in super-resolution tasks. However, the aforementioned image super-resolution models typically require a significant amount of computational resources and storage space. This is particularly problematic when handling high-resolution images, as they tend to consume excessive Graphics memory. This is an urgent challenge that needs to be addressed, and our research can effectively resolve this issue.

## A.2 IMPLEMENTATION DETAILS OF CLASSIFICATION

In our experiments, we continue the same experimental setup as Auglocal. The experiments on CIFAR-10 Krizhevsky et al. (2009), SVHN Netzer et al. (2011), and STL-10 Coates et al. (2011) datasets with ResNet-32He et al. (2016). We employ batch sizes of 1024 for CIFAR-10 and SVHN and 128 for STL-10. The training duration spans 400 epochs, starting with initial learning rates of 0.8 for CIFAR-10 / SVHN and 0.1 for STL-10, following a cosine annealing scheduler Coates et al. (2011).

## A.3 IMPLEMENTATION DETAILS OF OBJECT DETECTION

**Dataset:** To validate the model's ability to fit large datasets, we use the VOC detection datasetEveringham et al. (2010) containing 9,963 images and the COCO datasetLin et al. (2014)

Table 6: Comparison of GPU memory usage between BP and other methods on the CIFAR-10.

| Dataset | Network | Method | GPU Memory(GB) |
|---|---|---|---|
| CIFAR-10 | ResNet-32(K=16) | BP | 3.37 |
| | | **DGL** | **2.25(↓ 33.2%)** |
| | | **InfoPro** | **2.31(↓ 31.5%)** |
| | | **PredSim** | **1.91(↓ 43.3%)** |
| | | **Auglocal** | **1.67(↓ 50.4%)** |
| | | **Ours** | **1.77(↓ 47.5%)** |
| | ResNet-110(K=55) | BP | 9.26 |
| | | **DGL** | **2.44(↓ 73.7%)** |
| | | **InfoPro** | **2.38(↓ 74.3%)** |
| | | **PredSim** | **1.90(↓ 79.5%)** |
| | | **Auglocal** | **1.72(↓ 81.4%)** |
| | | **Ours** | **1.84(↓ 80.1%)** |

containing 123,287 images for our object detection experiments. Additionally, all backbones are pre-trained on the ImageNet dataset, which includes approximately 1.3 million images.

**Model Variants:** To validate the scalability of the proposed method, we employ entirely different network architectures, namely YOLO Redmon et al. (2016) and RetinaNet Lin et al. (2017c). For a fair comparison with other models, the YOLO model used ResNet-based YOLOv1. Networks using the local detection method are referred to as FBA versions. Each model was trained using ResNet-34, ResNet-50, ResNet-101, and ResNet-152.

Furthermore, to compare the performance of the local detection method with other local learning methods in terms of memory overhead reduction, we conduct comparisons under the same model partitioning conditions. We adopted the state-of-the-art local learning method DGL Belilovsky et al. (2019) for the object detection task. To validate the effectiveness of the local detection algorithm, we compared its memory-saving performance.

**Training and Fine-tuning:** We utilize the SGD optimizer Keskar & Socher (2017) with Nesterov momentum Dozat (2016) set at 0.9 and an L2 weight decay factor of 1e-4. The training duration spans 90 epochs, with a learning rate employing a warm-up strategy that is set to 0 for the first 5 iterations, followed by 1e-4, and adheres to a cosine annealing schedule. When using ResNet-34 as the backbone, it is divided into 16 modules. Similarly, when employing ResNet-50, ResNet-101, and ResNet-152 as backbones, the networks are divided into 16, 33, and 50 modules, respectively. This division is based on the block parameters used in the construction of ResNet, with each local module's auxiliary network having its unique parameters. During training, RetinaNet uses a batch size of 64, whereas YOLO uses a batch size of 32.

### A.4 IMPLEMENTATION DETAILS OF SUPER-RESOLUTION

**Dataset:** For the super-resolution task, we utilize the DIV2K Agustsson & Timofte (2017) dataset, which comprises over 1000 high-resolution images, each exceeding 2K in resolution. This dataset is extensively employed in various super-resolution challenges and competitions.

**Model Variants:** On the DIV2K dataset, we conduct tests for 2x, 3x, and 4x super-resolution tasks to evaluate the model's performance, using EDSR as the benchmark model. The configurations employing the FBA method for local learning are denoted as EDSR-FBA.

**Training and Fine-tuning:** During training, we use patches of 48x48 low-resolution (LR) images and their corresponding high-resolution (HR) patches. ADAM is used as the optimizer, with the learning rate set at 1e-4. Initially, we begin training from scratch on the ×2 model. Once the model converges, it is used as a pre-trained network for training on the ×3 and ×4 models.

### A.5 MORE RESULTS

**GPU Memory in Classification:** We compared the classification task with the current state-of-the-art Local learning task in terms of GPU Memory overhead. As shown in 6, our FDA approach is also the most advanced in reducing the memory overhead of the GPU Memory.

**Representation Similarity:** We conduct a Centered Kernel Alignment (CKA)Kornblith et al. (2019) experiment to validate the effectiveness of FBA. Specifically, we calculate the CKA similarity for each layer using FBA, DGLBelilovsky et al. (2019), and BP under different methods and averaged results. As shown in Figure 5, the representation differences among the methods are minimal in the later layers, with DGL being closer to BP than FBA. However, FBA achieves higher similarity in the early layers due to its Focal Features Selection, which aids the model in learning key discriminative features early on. This experiment confirmed that FBA's design enhances the model's understanding of early features.

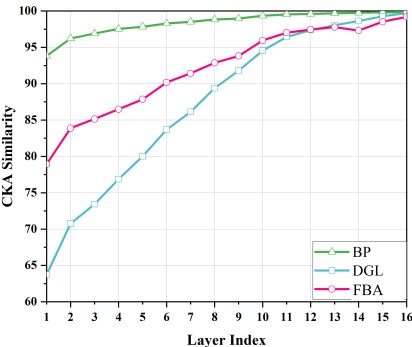

Figure 5: Diagram of the construction method of FBA

