# OpenReview forum: "Advancing Supervised Local Learning Beyond Classification with Long-term Feature Bank"
_ICLR.cc/2026/Conference — ICLR 2026 Conference Withdrawn Submission_

### Official Review · Reviewer_U5rV · 2025-10-17

**Soundness:** 3
**Presentation:** 3
**Contribution:** 2
**Rating:** 2
**Confidence:** 4

**Summary:**

Unlike previous local learning methods restricted to classification tasks, this work extends supervised local learning to a wide range of visual tasks. It introduces the Feature Bank Augmented (FBA) auxiliary network to enhance cross-task adaptability and feature communication. The design unifies and simplifies local learning architectures, making them more flexible and task-agnostic. As a result, FBA achieves comparable performance to end-to-end training while retaining the GPU memory efficiency of local learning.

**Strengths:**

The proposed FBA framework offers a unified and efficient solution that extends local learning to diverse visual tasks. By incorporating cross-scale feature access, it enhances representation quality and adaptability while maintaining a simplified architecture. Experiments demonstrate that FBA achieves performance comparable to end-to-end backpropagation with substantially lower GPU memory usage, highlighting its effectiveness and efficiency across various applications.

**Weaknesses:**

1) There are some typos in formulas and citations in the main text. For example, Eq (1) and citations in Page 4. Also there are irregular spacing, please check it. 2) I don't understand FPN existing in the main text. It'd be better to introduce relavant infomation. 3) Since the feature bank is one of the most contribution in your work, there should be more contents to explain that how the feature bank works in the three tasks or at least from a general level. After reading the main text, I just understand what are included in feature bank but don't understand how it works. 4) For experimental part, based on table 1 & 2, for example, the training time of 'ours' is more that 'DGL'; and in table 2, the performance seems vey close between 'ours' and resnet models. I did not see a obvious advantage of 'ours' compared to SOTA methods. Combining with the table 3, while your method offers improvements in GPU memory efficiency, it provides no substantial benefits in performance and training time. 5) From the method level, I think the novelty is not enough for iclr. From my point of view, your framework introduces a fine-grained simple local network combining with a feature bank. Also it did not lead a significant improvement on performance and training time. If authors aim to save more memory not on improving performance, they should include corresponding experiments to support it. Also I did not see the advantages on training time.
Also some of weaknesses like typos/errors have happened in the 2025iclr submission, but the author still did not correct them and submit to 2026iclr again.

**Questions:**

I have no other questions. Please review the weakness part in my review. That's the most important contents to me.

---

### Official Review · Reviewer_TMRP · 2025-10-28

**Soundness:** 2
**Presentation:** 1
**Contribution:** 2
**Rating:** 2
**Confidence:** 4

**Summary:**

This paper aims to address the limitation of local learning which is restricted to image classification tasks by introducing a novel framework called Feature Bank Augmented auxiliary network (FBA). The proposed FBA network utilizes multi-scale and task-critical features which allows information flow across isolated modules and mitigates the “short-sightedness” issue common in local learning. With another proposed Simple Local Modules (SLM), a lightweight, task-agnostic auxiliary networks aligned with backbone stages, the FBA enables modular local updates and easily to be expanded to different down-stream tasks. Experimental results show some improvement on classification, and marginal improvement on object detection and super-resolution.

**Strengths:**

1. This paper aims to address the limitation of local learning that restricted to image classification tasks, which is a valuable motivation.
2. The proposed method helps to reduce the GPU memory consumption during training process according to the experimental results.

**Weaknesses:**

1. The document is not well-organized. This paper is like a draft and the reader needs to take a lot of effort to understand the main pipeline of the proposed framework. There are a lot of inappropriate inline citations separating the sentences illogically. A lot of subscripts of the formula is directly put on the baseline following the main-text instead of below the base line.

2. The proposed method is less innovative. The AugLocal[1] also utilizes the features from cnn layers and fc layers of the network. The proposed feature bank, i.e., cached fc layer activations, is more like a technical optimization. In Table 3, the only-one ablation study is insufficient to show the effectiveness of the feature bank.

3. In Table 1, the reported numbers of SOTA methods are from the original papers, which is not a fair experimental settings, while the proposed method only marginal outperforms the most recent method AugLocal. The training time is not obviously decreased on CIFAR-10, STL-10, and SVHN. In Table 6, the proposed method has a higher GPU memory consumption compared to AugLocal.

4. In Table 2 and Table 3, for object detection and super-resolution, it lacks comparison to the sota methods. Why the proposed method is only compared to DGL with RetinaNet-R34 backbone?

[1] ChenxiangMa, JibinWu, Chenyang Si, and KayChen Tan. Scaling supervised local learning with augmented auxiliary networks. arXiv preprint arXiv:2402.17318,2024.

**Questions:**

Please see the weaknesses.

1. I recommend authors to provide more results on the same training settings and compare to sota methods on image classification task.

2. I recommend authors to provide more (if possible) experimental results of sota methods on object detection / super-resolution.

3. More ablation study on the proposed feature bank is recommended to illustrate the effectiveness of the motivation.

---

### Official Review · Reviewer_gzDY · 2025-10-31

**Soundness:** 2
**Presentation:** 2
**Contribution:** 2
**Rating:** 4
**Confidence:** 3

**Summary:**

The article proposes a local-learning method called FBA, which aims to address the lack of task adaptability and insufficient cross-scale feature communication in existing auxiliary network architectures. The experimental results show that FBA maintains BP-level performance across different tasks by leveraging the proposed feature bank and cross-scale feature utilization.

**Strengths:**

1. The paper proposes FBA to improve task adaptability and cross-scale feature communication in auxiliary networks.
2. Experiments show that FBA maintains BP-level performance across multiple tasks.
3. The figures are clear with pleasing color design.

**Weaknesses:**

1. In Figure 3, it appears to illustrate how the proposed method constructs the auxiliary network when the same model is applied to different tasks. However, the color and structure of the SLM modules are not very clear. The meaning of the color variations in the feature layers and their correspondence to elements within the SLM is ambiguous. For example, in the upper subfigure, the color order of the first SLM is reversed, and in the lower subfigure, the last three SLMs share the same color. No detailed explanation is provided in the figure or the main text.

2. There are several issues related to wording and formatting. For instance, the description of "Simplify" in line 258 refers to (3). In the appendix, the term "the most advanced" in line 755 is inconsistent with the results in Table 6.

**Questions:**

During training, is the feature bank updated online in real time, or is it precomputed before training and reused throughout the process?

---

### Official Review · Reviewer_cKwt · 2025-11-01

**Soundness:** 2
**Presentation:** 2
**Contribution:** 2
**Rating:** 2
**Confidence:** 4

**Summary:**

This paper proposes the Feature Bank Augmented auxiliary network (FBA), a novel framework designed to extend supervised local learning beyond its traditional application in image classification. The authors identify that existing local learning methods suffer from poor cross-task adaptability and a "short-sightedness" problem caused by a lack of cross-scale feature communication. FBA addresses these limitations with two main contributions: 1) Simple Local Modules (SLM), which provide a simplified, adaptable design for the auxiliary networks, and 2) a Feature Bank that captures and shares critical multi-scale features with the local modules during training to improve communication. The paper's primary contribution is demonstrating that this task-agnostic approach achieves performance comparable to end-to-end back-propagation on diverse tasks, including object detection and super-resolution, while significantly reducing GPU memory usage

**Strengths:**

1: The proposed local learning framework can achieve performance comparable to end-to-end backpropagation on complex, non-classification tasks like object detection and super-resolution, while substantially reducing GPU memory consumption.

2: The Simple Local Module (SLM) aims to address the challenge of cross-task adaptability by reusing aligned backbone blocks and the original task head, removing the need for complex, manual, task-specific auxiliary networks.

3: Extensive experiments on various tasks and model architectures show promising results in using local learning and a long-term feature bank to reduce the GPU memory consumption.

**Weaknesses:**

1: **The presentation needs significant improvements for publishing.** It's very hard to catch the main ideas from Figures 2, 3, and 4. The overall framework seems to just be reusing the feature of the backbone or any internal blocks.

2: **The paper has limited contributions.** The paper's primary motivation for replacing end-to-end BP is not sufficiently argued; while it mentions BP's limitations like memory usage, the necessity of a local-learning alternative versus the performance trade-offs is not deeply explored. Furthermore, the core technical contributions, the Simple Local Module (SLM) and the Feature Bank, appear incremental, which has also been widely explored, such as dense connections and incremental learning for classification and object detection.

3: **Potential Overclaiming.** The paper claims to be the "first task-agnostic framework that extends supervised local learning beyond classification". This may be an overstatement, as prior works (e.g., InfoPro) have already been successfully extended to dense prediction tasks like segmentation, a point not adequately addressed by the authors.

4: **Evaluation is limited.** The paper's claims of generalizability are not fully supported by its experiments. First, the evaluation is restricted entirely to the image domain, with no discussion of how this framework might apply to other modalities (e.g., language, multimodal tasks). Second, and more critically, within the visual domain, the experiments are exclusively conducted on backbones with inherent multi-scale feature hierarchies (e.g., ResNet). There is no evaluation or discussion of how FBA would be instantiated with non-pyramidal architectures like Vision Transformers (ViT) or DETR.

**Questions:**

1: The paper presents FBA as a "task-agnostic" framework that "eliminates manual configuration". However, the implementation of the Feature Bank appears to be highly task-specific: it is designed to store GAP/FC activations for classification, multi-scale maps to build a local FPN for detection, and early-stage high-resolution maps for super-resolution. Could the authors clarify this apparent contradiction?

2: There is a clear discrepancy in performance between different tasks. While FBA "consistently matched or slightly outperformed" back-propagation on object detection (Table 3) , it shows a consistent and non-trivial performance drop on super-resolution (e.g., -0.73dB for x2, -1.71dB for x3, and -1.21dB for x4 in Table 5). Could the authors elaborate on the underlying reasons for this performance gap?

---

### Note · Authors · 2025-11-17

I have read and agree with the venue's withdrawal policy on behalf of myself and my co-authors.